# Advances of Wnt Signalling Pathway in Colorectal Cancer

**DOI:** 10.3390/cells12030447

**Published:** 2023-01-30

**Authors:** Yaoyao Zhu, Xia Li

**Affiliations:** 1Marine College, Shandong University, Weihai 264200, China; 2Shandong Kelun Pharmaceutical Co., Ltd., Binzhou 256600, China

**Keywords:** Wnt signalling pathway, colorectal cancer, invasion and metastasis, colitis-associated carcinoma, cancer stem cells

## Abstract

Colorectal cancer (CRC) represents one of the most common cancers worldwide, with a high mortality rate despite the decreasing incidence and new diagnostic and therapeutic strategies. CRC arises from both epidemiologic and molecular backgrounds. In addition to hereditary factor and genetic mutations, the strongly varying incidence of CRC is closely linked to chronic inflammatory disorders of the intestine and terrible dietary habits. The Wnt signalling pathway is a complex regulatory network that is implicated in many CRC physiological processes, including cancer occurrence, development, prognosis, invasion, and metastasis. It is currently believed to include classical Wnt/β-catenin, Wnt/PCP, and Wnt/Ca^2+^. In this review, we summarise the recent mechanisms and potential regulators of the three branches of the Wnt signalling pathway in CRC.

## 1. Introduction

Wnt signalling is one of the key cascades regulating normal development and physiology and has also been tightly evolved to perform diverse functions in cancer [1,2]. Wnt signal transduction is divided into the Wnt/β-catenin or canonical Wnt signalling pathway, the planar cell polarity (Wnt/PCP) pathway, and the Wnt/Ca^2+^ signalling pathway [3]. Abnormal Wnt pathway components have been identified as drivers of cancer and as potential targets for cancer treatment, especially in colorectal cancer (CRC) [4]. CRC is the third most common cancer worldwide [5,6]. In recent years, the majority of CRCs have been sporadic and early-onset, which is largely attributed to a constellation of modifiable environmental risk factors (for example, obesity, physical inactivity, poor diet, alcohol consumption, and smoking) [7,8,9]. It is now widely accepted that the Wnt signalling pathway plays a pivotal role in tumour development, progression, invasion, and metastasis, and in drug resistance and the relapse of cancer. In this review, we determine the impact of the canonical and non-canonical Wnt signalling pathways in driving the occurrence, growth, invasion, and chemoresistance of CRC tumours, as well as their role in prognosis, to comprehensively understand the role enacted by the Wnt pathway in CRC [10].

## 2. The Canonical Wnt/β-Catenin Pathway

The Wnt/β-catenin signalling pathway is associated with physiological processes and contributes to the diverse processes of some solid tumours and haematological malignancies, such as CRC [11,12]. Under normal physiological conditions, the transcription factor β-catenin, the crucial component of the Wnt signalling pathway, is destroyed by the β-catenin destruction complex formed by adenomatous polyposis (APC), casein kinase I (CK I), glycogen synthase kinase 3β (GSK3β) and Axin in which Axin is a scaffolding protein, and CK I and GSK3β phosphorylate β-catenin in turn [13,14,15]. Subsequently, the phosphorylated β-catenin is recognised and degraded by E3 ubiquitin ligase (β-TrCP) [16]. Low-density lipoprotein receptor-related protein 5/6 (LRP5/6), co-receptors for Wnt ligands, are associated with Frizzled (FZD) receptors. Once bound by Wnt, FZD/LRP complex activates the canonical signalling pathway. The Dishevelled (Dvl), a cytoplasmic protein that functions as upstream of β-catenin and GSK3β, is activated. Dvl is required for the phosphorylation of LRP5/6 PPPSPxS motifs. LRP5/6 are phosphorylated by several protein kinases including GSK3β and CK I, and recruits the scaffold protein Axin. Multimers of receptor-bound Dvl and Axin molecules support the formation of the LRP-FZD dimer. The loss of function of the destruction complex leads to the stabilization of β-catenin. As a result, β-catenin accumulates in the cytoplasm, undergoing nuclear translocation, and activates the transcription factor T cell factor (TCF)/lymph enhancer factor (LEF) family to initiate the expression of Wnt target genes [17,18,19,20].

## 3. The Non-Canonical Wnt Signalling Pathway

In addition to the Wnt/β-catenin signalling pathway, many researchers have indicated that the non-canonical Wnt signalling pathway also has a vital role, with growing links to cancer progression, exerting both oncogenic and tumour-suppressive effects [21]. The main non-canonical Wnt signalling pathways are the Wnt/PCP and Wnt/Ca^2+^ signalling pathways [22]. Wnt1, Wnt2, Wnt3, Wnt3a, Wnt8a, Wnt8b, Wnt10a, and Wnt10b are regarded as canonical Wnt proteins, while Wnt4, Wnt5a, Wnt5b, Wnt7a, Wnt7b, and Wnt11 belong to the non-canonical Wnt proteins, which initiate canonical and non-canonical pathways, respectively, by activating Frizzled receptors (Figure 1).

### 3.1. The Wnt/PCP Signalling Pathway

The polarization of many epithelia within the plane of the epithelium is referred to as planar cell polarity (PCP), and the associated molecular pathway can regulate cellular polarity in non-epithelial/epithelial contexts [23]. PCP is an evolutionarily conserved pathway that is regulated by Frizzled (FZD)/PCP core group, which includes the seven-pass transmembrane receptor Frizzled (FZD), the atypical cadherin Flamingo (Fmi/Celsr), the tetraspanin-like transmembrane protein Van Gogh (Vang or Stbm/Vangl), and the cytosolic components Dishevelled (Dsh/Dvl), Diego (Dgo/Ankrd6), and Prickle (Pk) [24]. Wnt5a-Frizzled binding needs to transduce signals with a co-receptor Ror2 and induces the formation of a Vangl2-Ror2 complex [25]. In addition, phosphorylated Vangl contributes to Vangl2-Ror2 localisation and/or function, which requires the activity of CK I and Dvl [26]. Downstream signalling cascades (RHO-ROCK (Rho-associated kinase) and RAC-JNK (c-Jun-N-terminal kinase)) in the Wnt/PCP pathway are activated [27]. Then, RHO, RAC, JNK, and ROCK activate a downstream target gene, which is involved in neuronal plasticity, regeneration, cell death, and the regulation of cellular senescence [28].

### 3.2. The Wnt/Ca^2+^ Signalling Pathway

More and more evidence indicates that the Wnt/Ca^2+^ signalling pathway influences cell fate decisions and cell migration [29]. Wnt-Fzd receptor ligand interaction along with the phosphorylation of co-receptor Ror by GSK3β activates phospholipase C, which derives inositol 1,4,5-triphosphate (IP3) and 1,2 diacylglycerol (DAG) from phospholipid phosphatidyl inositol 4,5-bisphosphate [30]. IP3 regulates the endoplasmic reticulum (ER) by releasing calcium ions, resulting in the activation of calcium calmodulin-dependent protein kinase II (CaMKII) and phosphatase calcinurin (Cn). Together, DAG and the calcium ions released from the ER activate protein kinase C (PKC) [31]. Both CaMKII and PKC activate various regulatory proteins (NF-κB and CREB), and Cn can activate the cytoplasmic protein nuclear factor associated with T cells (NFAT) [32]. NFAT, NF-κB, and CREB translocate to the nucleus and transcribe downstream regulatory genes [33]. The Wnt/FZD receptor ligand interaction may also activate phosphodiesterase 6 (PDE6) in a calcium-dependent manner, leading to a decrease in cyclic guanosine monophosphate (cGMP). 

## 4. The Wnt Signalling Pathway in Occurrence of Colorectal Cancer (CRC)

The Wnt signalling pathway is one of the most representative signalling pathways in CRC, playing an important role in the occurrence of CRC (Figure 2). According TCGA data, in colorectal cancer (CRC; *n* = 594), using Memorial Sloan Kettering Cancer Center (MSKCC), the largest fraction of sporadic tumours accumulates APC mutations (67%), followed by lower fractions of ring finger protein 43 (RNF43) (8%), β-catenin (6%), and Axin2 (5%) mutations according to frequency of mutation or deep deletion of the Wnt pathway genes [34]. In addition to the Wnt/β-catenin signalling pathway, many researchers have indicated that the non-canonical Wnt signalling pathway also has a vital role, with growing links to cancer progression, exerting both oncogenic and tumour-suppressive effects [21]. The main non-canonical Wnt signalling pathways are the Wnt/PCP and Wnt/Ca^2+^ signalling pathways [22]. 

### 4.1. The Wnt Signalling Pathway and Environmental Exposure 

CRC is divided into sporadic and familial/hereditary cases, emphasising an aetiology with shared environmental factors and genes. Driver mutations in APC are representative of CRC subsets with distinct characteristics. Inactivating APC mutations causes familial adenomatous polyposis (FAP), which is a syndrome characterized by extensive intestinal polyp formation and a high risk of CRC development [35]. Moreover, a large number of APC mutations have also been found in sporadic CRCs [36]. In addition to individual and family genetic factors, the incidence of sporadic CRC (sCRC) and early colon cancer is increasingly common due to unhealthy lifestyles and environmental exposure [37]. 

#### 4.1.1. Lifestyle

The causes of CRC tumorigenesis and progression are complicated and may include complex interactions. Apart from heredity factors, an unhealthy lifestyle is regarded as a potential risk factor associated with CRC. Diet influences a large part of the maintenance and function of the colon environment in mammals, and biological changes may be associated with the incidence of CRC [38,39]. A recent study indicated that linoleic acid (LA)-enriched diets promote chemically induced CRC via the Wnt/β-catenin signalling pathway. The 15-LOX-1 peroxidation of LA in phosphatidylinositol-3-phosphates (PI3P_LA) leads to the formation of PI3P_13-HODE, which decreases PI3P binding to SNX17 and LRP5 and inhibits the recycling of LRP5 from endosomes to the plasma membrane, thereby increasing LRP5 lysosomal degradation [40]. Moreover, the effect of a high-fat diet on the intestinal stem cell (ISC) niche is related to CRC risk: a high-fat diet resulted in the dysregulation of colonic mesenchymal stromal cells (MSCs), including increased cell numbers, increased bile acid (BA) receptor expression on colonic MSCs, the overexpression of Wnt2b, and the activation of CAF-like properties, which promoted crypt regeneration. In this way, BAs stimulate colonic MSC-induced Wnt2b production through the activation of FXR signalling [41]. 

Furthermore, a recent study found that the circadian clock was seemingly regarded as a risk factor impinging on intestinal physiology [42]. The molecular clock consists of the core transcription factors, circadian locomotor output cycles kaput (CLOCK), and aryl hydrocarbon receptor nuclear translocator-like protein 1 (Arntl encoding BMAL1), which regulate the clock gene expression programmes [43]. In human colorectal tumours, clock gene expression disorders cause abnormal cell cycle progression and regulate Wnt secretion in the intestinal epithelium [44]. Recently, researchers found the molecular pathways linking circadian disruption and the pathogenesis of CRC by developing a new tissue-specific genetically engineered mouse model (GEMM). The results demonstrated that the environmental disruption of the circadian clock accelerated the tumour burden in the APC mutant GEMM, as well as patient-derived organoids (PDOs), which highlights the importance of the circadian clock in the intestine. In addition, data demonstrated that disruption of intestinal Bmal1 accelerated the heterozygosity (LOH) of APC, and Wnt/β-catenin signalling was aberrantly activated, enhancing MYC-dependent glycolytic metabolism to accelerate CRC progression [45].

#### 4.1.2. Pro-Tumorigenic Bacteria

The involvement of gut microbiota in CRC carcinogenesis has been extensively studied. Current evidence suggests that bacteria contribute to the development of CRC, partly by stimulating chronic inflammation [46,47]. Colonisation with anti-virulence factor A (AvrA) expressing *Salmonella enterica* significantly reduced several proteins of the Wnt family, including Wnt1, Wnt2, Wnt3, Wnt6, Wnt9a, and Wnt11 [48,49]. Furthermore, exposure to a short-term colibactin-producing (pks+) *Escherichia coli* infected healthy murine colon epithelial cells, which led to mutations in the Wnt signalling pathway in a study utilising organoids [50]. In terms of *Fusobacterium nucleatum* in CRC carcinogenesis via the Wnt signalling pathway, it has been reported that the *F. nucleatum*-associated CRC process was modulated by the phospholipid-binding protein Annexin A1, which was upregulated in cancerous cells by binding FadA, a *F. nucleatum* adhesion molecule, to E-cadherin, resulting in the activation of β-catenin-regulated transcription and an increased expression of transcription factors and the oncogenes c-Myc and Cyclin D1 [51,52]. Some researchers also believe that *F. nucleatum* activates the Wnt/β-catenin pathway by the upregulation of cyclin-dependent kinase 5 (Cdk5) to promote the proliferation and migration of cells [53]. Another study reported that *F. nucleatum* stably adheres to cancer stem cells (CSCs), likely by multiple interactions involving the tumour-associated Gal-GalNac disaccharide and the Fn-docking protein CEA-family cell adhesion molecule 1 (CEACAM-1), contributing to microbiota-driven colorectal carcinogenesis [54]. 

#### 4.1.3. Gut Microbiota

The alteration of gut microbiota in all mucosal surfaces of the colon causes dysbiosis of the gut environment, which is involved in CRC tumorigenesis [55]. Intestinal microbiota activate the pathogen recognition receptors PRR9 to induce innate immune responses and activate the inflammatory response mediated by transcription factor IRF3 and NF-κB [56]. Recently, the author established the relationship between cytosolic IRF3 and Wnt/β-catenin signalling via AOM/DSS-induced chronic colonic inflammation mice models, demonstrating that IRF3 is a tumour suppressor, inhibiting Wnt signalling, instead of functioning as a transcription factor. In normal physiology, the site of a.a. 634–663 in the exact armadillo repeats (ARM) of β-catenin links IRF3-β-catenin interaction, which retains β-catenin in the cytoplasm and inhibits the nuclear import of β-catenin. Gut microbiota trigger IRF3 phosphorylation and the structural/interface change of activated IRF3, resulting in IRF3 failing to bind β-catenin or active-β-catenin (β-catenin-S33A), thus losing its capacity to retain β-catenin in the cytoplasm. The IRF3-β-catenin axis links the gut microbiota to the proliferation of the intestinal epithelium and the development of CRC [57].

### 4.2. The Wnt Signalling Pathway in Colitis-Associated Carcinoma

Apart from sporadic CRC, colitis-associated CRC (CAC) can arise and has caught the attention of the public. A growing number of studies show that intestinal inflammation drives the initiation of cancer and its development [58]. Inflammatory bowel disease (IBD), including Crohn’s disease (CD) and ulcerative colitis (UC), is a chronic, idiopathic, relapsing inflammatory disease of the gastrointestinal (GI) tract [59]. Chronic and long-term IBD changes are prone to turn to colonic carcinoma. Dysregulation of the Wnt/β-catenin pathway, such as the GSK3β and APC gene, plays a vital role in CAC progression [60,61,62].

Inflammation-related pathways are activated in the early stages of CAC, such as NF-κB signalling [63]. Importantly, Wnt/β-catenin signalling could be activated as a key regulator of the CRC transformation of CAC [64]. The differences in IBD-CRC formation concern the loss of APC and p53 function. Compared with p53, loss of function occurs early in CAC carcinogenesis, loss of APC function is less frequent and takes place later in the development of malignancy [65,66]. The author characterised IBD-associated tumorigenesis by multiple high-throughput approaches and compared the results with in-house data sets from sCRCs. The results demonstrated that Wnt signalling dominated the mesenchymal stroma-rich subtype instead of the canonical epithelial tumour subtype in IBD-CRCs, which may affect prognosis and treatment options. Negative Wnt regulators Axin2 and RNF43 were strongly downregulated in IBD-CRCs and chromosomal gains at HNF4A, a negative regulator of Wnt-induced epithelial–mesenchymal transition (EMT), were less, which compared to sCRCs, while, polymeric immunoglobulin receptor (PIGR) and Oncostatin M Receptor (OSMR) were dysregulated via epigenetic modifications., which is involved in mucosal immunity. Therefore, the mechanism of Wnt pathway dysregulation indicated that IBD-CRCs toward mesenchymal tumour subtype instead of canonical epithelial tumour subtype. Meanwhile, somatic mutations in APC and Kirsten rat sarcoma viral oncogene (KRAS) were less frequent in IBD-CRCs compared to sCRCs [67]. Recently, research found the mechanism responsible for the crosstalk between NF-κB and Wnt/β-catenin signalling. Dvl2, an adaptor protein of Wnt/β-catenin signalling, interacts with the C-terminus of tumour necrosis factor receptor 1 (TNFR1) and mediates TNFR1 endocytosis, leading to NF-κB signal inhibition and promoting CAC progression. In addition, STAT6 negatively regulates Dvl2 levels and restrains cancer cell colony formation by interleukin- 13 (IL-13) [68].

Intestinal mucosal injury is an important pathological change in individuals with IBD [69]. Normal cell turnover of IECs provides the foundation to guarantee intestinal epithelial homeostasis; excessive IEC death is a well-known clinical hallmark of IBD [70]. The Wnt/β-catenin signalling pathway is highly conserved during evolution and is involved in other signalling pathways, such as BMP signalling, to drive the orderly self-renewal and differentiation of ISCs and maintain intestinal homeostasis [71]. c-Myc and Cyclin D1, the target genes of the Wnt signalling pathway, promote the proliferation of intestinal epithelial cells (IECs), enhancing the activity of ISCs [72]. Recent study has indicated that IL-36γ upregulated extracellular matrix and cell-matrix adhesion molecules and positively correlated with beta-catenin levels facilitated by Wnt signalling. Moreover, the activation Wnt signalling is mitigated by IL-36Ra or IL-36γ neutralising antibody, in which IL-36γ and IL-36Ra played the critical roles of in gut inflammation and tumorigenesis through Wnt/β-catenin [73]. MCL1, a member of the pro-survival BCL2 family, has been identified as having an essential role in the preservation of intestinal homeostasis and the prevention of carcinoma [74]. IEC-specific MCL1 deficiency shares hallmark features with IBD and is independent of microbe-induced chronic inflammation. Further investigation into it functions revealed strong γH2AX positivity and GSEA (from RNASeq data), indicating that MCL1 deletions induce DNA damage and abnormal Wnt signalling activation, with a marked increase in Wnt2b expression levels in the stroma surrounding the intestinal crypts, which promotes a clear proliferation-enhancing effect [75].

Wnt/β-catenin signalling axis in regulatory T cells promotes IBD and colonic dysplasia. Treg cells are crucial in response to the inflammatory milieu, RORγt is one of the transcription factors for multiple Treg cell subpopulations are described that express additional helper T (TH) cell lineage [76,77]. In regulatory T (Treg) cells, excessive Wnt/β-catenin activation may impact their function due to that TCF1 is known the bona fide DNA-binding partner of β-catenin [78]. Foxp3 binding to gene enhancers made accessible by Foxo1 regulates a core regulatory programme of late differentiation of Treg cells to define Treg cells [79]. The overlap of Foxp3 binding with the transcription factor TCF1 impacts Treg cell function [80]. Recent findings indicated that TCF1 and Foxp3 together limited the expression of pro-inflammatory genes in Treg cells, which function was interfered by activation of β-catenin signalling interferes with this function, and activation of β-catenin signalling promoted the disease-associated RORγ t(+) T-reg phenotype [81].

Apart from individual gene mutations and familial hereditary factors, hazardous unhealthy lifestyle, such as diet and disorder circadian clock, pro-tumorigenic bacteria and imbalanced gut microbiota, may cause imbalance of Wnt signalling pathway and increase the rates of occurrence of CRC, especially sporadic and early-onset CRC. At the same time, the role of Wnt signalling pathway in colitis-associated carcinoma should not be overlooked. Wnt signalling pathway has been found to have differences between sCRC and colitis-associated carcinoma.

## 5. The Wnt Signalling Pathway in the Development of CRC

### 5.1. The Wnt Signalling Pathway in Cancer Stem Cells (CSCs)

CSCs with pluripotency and self-renewal properties are considered responsible for the progression and recurrence of the disease and for tumour resistance [82,83,84]. A rapid turnover rate maintains intestinal homeostasis by the asymmetric division and proliferation of SCs at the base of the intestinal crypts, migration, and differentiation of the SCs at intestinal villus [85]. Imbalances in the signalling between the microenvironment and ISCs play a key role in promoting the growth and progression of established CRCs [82]. The Wnt signalling pathways participate in CRC survival, proliferation, and self-renewal properties, and most of them are dysregulated in CSCs [86]. Canonical Wnt signalling causes the expansion of rapidly cycling CSCs and modulates both immune surveillance and immune tolerance, while non-canonical Wnt signalling supports the maintenance of slow cycling to promote epithelial–mesenchymal transition and immune evasion via crosstalk with TGFβ (transforming growth factor-β) signalling cascades [87]. Wnt-related cancer stemness features relating to proliferation/dormancy plasticity will be discussed here, while other cancer stemness features will be discussed in the following section.

ISC dynamics are influenced by many endogenous and exogenous stimuli, including, for example, diet and the presence of niche factors. Leucine-rich repeat-containing G-protein coupled receptor 5 (Lgr5), the first functional relation marker of ISC features, is the Wnt target gene. Lgr5^+^ cells, which constitute an important component of the epithelial stem cell niche, are capable of long-term self-renewal and possess multilineage differentiation potential. The key signalling pathway controlling the ISC function is the Wnt/β-catenin pathway, and the Wnt and R-spondin ligands are extensively secreted by the niche, reaching their highest levels in the bottom of the crypt [88,89,90,91]. Although the Hippo transcriptional coactivator YAP is considered oncogenic in many tissues, its roles in intestinal homeostasis and CRC remain controversial. A study confirmed that YAP reprogrammes Lgr5^+^ proliferating CSCs into Lgr5- dormant CSCs, even if an Apc mutation via CRISPR-Cas9 into tetO-YAP rtTA colon organoids present constitutively active Wnt signalling pathway [92].

Wnt pathways are significant in terms of orchestrating intestinal stem cell and differentiation cues. SOX9, a transcriptional activator, is a key early mediator of CRC due to it binding to enhancers that upregulate genes associated with Paneth and stem cell function. SOX9 directly activates PROM1 using a Wnt/TCF4-responsive intronic enhancer to impair intestinal differentiation and promote CRC growth by a PROM1-SOX9 positive feedback loop [93]. ISCs play a crucial role in renewing the intestinal epithelium and maintaining tissue homeostasis [94]. Lgr5^+^ cells serve ISCs for the homeostatic balance of the intestine and can be a key driver of CSCs in intestinal tumorigenesis [95]. Many investigations have revealed that the Wnt signalling pathway contributes to the process of ISCs and CSCs; for example, β-catenin in ISCs is activated to promote the development of CRC due to a deficiency of APC [96]. Recently, researchers found the machinery for β-catenin activation using in vivo and in vitro experiments: the kinase MST4 phosphorylates β-catenin at Thr40 resulting in that β-catenin negatively influenced phosphorylation at Ser33 by GSK3β mediation. Subsequently beta-catenin binding to and being degraded by beta-TrCP was hindered, which causes subsequent accumulation and full activation of β-catenin, leading to increased numbers of ISCs/CSCs and exacerbated tumorigenesis [97].

Interestingly, some findings were conflicting for TCF4, especially a potential tumour suppressive role of the gene in intestinal cancer cells or tumours. Pathological transformation of the gut epithelium is driven by stabilized β-catenin by activation of Wnt/β-catenin, in which TCF4 is the major nuclear mediator. TCF4 was completely knockout in the adult colon that resulted in the formation of aberrant crypt foci (ACF), the earliest neoplastic lesions during CRC initiation [98]. A recent study reported the importance of mouse TCF4, mainly in adult intestinal epithelium homeostasis and intestinal tumour initiation. In contrast, in human cells, the TCF4 function is substituted by other LEF/TCF family members [99]. 

### 5.2. The Wnt Signalling Pathway in the Tumour Microenvironment

The tumour microenvironment (TME) is primarily composed of fibroblasts, immune cells, blood vessels, and extracellular matrix (ECM). Intercellular communications between tumour cells and the TME are mainly established through paracrine signalling, in which the Wnt/β-catenin pathway could be mediated by external factors to affect CRC progression [100]. Extracellular vesicles (EV) can transport mutant β-catenin and activate the Wnt signalling pathway in the recipient cells, thus promoting CRC progression. A recent study confirmed that Wnt2 acts in an autocrine manner, generating morphogenetic changes in fibroblasts and contributing to the invasive and metastatic capacity of CRC-derived cells and EVs play a key role in activating Wnt signalling in CRC [101]. In the solid tumour TME, hypoxia is a common effect, which may be mediated by β-catenin. Hypoxic CRC cells have been found to secrete exosomes enriched with Wnt4 ligands, resulting in the activation of β-catenin signalling in normoxic CRC cells to stimulate prometastatic behaviours, such as cell migration and invasion [102,103]. It has also been reported that Wnt5a has both tumour promoting and suppressing functions in CRC [104,105]. Researchers found that Wnt5a was overexpressed in tumour stroma in vivo and in vitro, especially M2-like tumour-associated macrophages (TAMs), and Wnt5a induced M2 polarisation of TAMs by regulating CaKMII-ERK1/2-STAT3 pathway-mediated IL-10 secretion [105], enhancing tumour growth, invasion and metastasis of CRC, and recruited macrophages whose infiltration depended on CaMKII-ERK pathway-mediated CCL2 secretion [106].

### 5.3. The Wnt Signalling Pathway in Angiogenesis

Angiogenesis is one of the key mechanisms of tumour development and is critical for invasive tumour growth, metastasis [107]. It has long been suggested that the Wnt/FZD/β-catenin signalling pathway regulates the development of blood vessels in physiological and pathological conditions. Sprouting angiogenesis is an essential condition for tumour growth and metastasis, to access the blood system in order to maintain sufficient oxygen and nutrients [108,109]. Tumour angiogenesis is mediated by tumour cells, cancer-associated fibroblasts (CAFs), and immune cells in the tumour stroma, and by breaking the balance of pro- and anti-angiogenic factors [110]. A high expression of Wnt2 was found in stromal CAFs and in the autocrine activation of canonical Wnt signalling and enhanced fibroblast motility [111]. Subsequently, researchers found that Wnt2 derived from stromal CRC-CAFs enhances angiogenesis by increasing extracellular migration and invasion and by altering the CAF secretome towards pro-angiogenic factors and ECM remodelling signals [112]. Long non-coding RNA (lncRNA) GAS5 inhibited the activation of the Wnt/β-catenin signalling pathway, thereby suppressing the angiogenesis, invasion, and metastasis of CRC [113]. TRIM24 was highly expressed in CRC tissues compared to nonneoplastic adjacent tissues; the overexpression of TRIM24 upregulates the expression of vascular endothelial growth factor (VEGF), CCL2/5, and CSF-1 via Wnt/β-catenin signalling to stimulate angiogenesis and recruited TAMs, which promotes CRC tumour growth [114].

Wnt4, a member of the Wnt family, has been reported in wound healing, acute kidney injury, and angiogenesis and may play a role in CRC via the β-catenin- relevance Wnt pathway. A recent study found that Wnt4 is highly express in the serum of CRC patients and in tumour tissues. Wnt4 prompted the nuclear translocation of β-catenin and increased the expression of α-SMA and fibronectin in NFs, which triggered the transformation of NFs into CAFs and enhanced reinforced cell contraction and microsphere formation. Meanwhile, ANG2, a critical gene for angiogenesis, could be abnormally activated by the Wnt4/β-catenin/ANG2 pathway, resulting in enhanced angiogenesis in CRC. Wnt4 may also be secreted as exosomes from CRC cells, playing a crucial role in promoting tumour progression in the TME [115]. Exosomes are nano-sized membrane vesicles with diameters between 30–100 nm and are generated from endosomal compartment invaginations [116]. CRC cell-derived exosomes have important roles in tumour progression through effectively delivering microRNAs, mRNAs, and proteins [117,118,119]. The protein levels of Wnt4 were higher in hypoxic exosomes than in exosomes derived from normoxic cells. In hypoxia, Wnt4-loaded exosomes secreted by the tumour cells promoted angiogenesis through the proliferation and migration of ECs via activated Wnt/β-catenin signalling [103,120]. 

The Wnt signalling pathway plays crucial effect on cancer stem cells, tumour microenvironment, and angiogenesis to drive the development of CRC. The complex microenvironment components, such as EV and exosomes, also contribute to development of CRC via Wnt signalling pathway.

## 6. The Wnt Signalling Pathway in the Invasion and Metastasis of CRC

CRC is a heterogeneous disease and lethality is especially common in the subtype of CRC with high stromal infiltration. As metastasis is a main reason for cancer-related deaths, and an aberrant Wnt/β-catenin signalling pathway causes CRC invasion and metastasis, it is therefore significant to understand the effect of the Wnt signalling pathway in invasion and metastasis in CRC [121]. In the nucleus, together with TCF/LEF, β-catenin binds to the promoter region of a target gene, and activates its transcription such as Twist and Slug, as E-cadherin transcription factors, which regulate EMT [122,123].

During invasive and metastatic cancer development, both the signalling and gene expression of cells to neighbouring cells and the coordination between cell–cell adhesion is changed. β-catenin acts as an adhesion linker for cadherin transmembrane receptors to the cytoskeleton. L1CAM (L1) cell adhesion receptors, a downstream target oncogene of β-catenin-TCF, were detected as regulators of invasive and metastatic CRC cells [124,125,126]. RNF128 inhibited cell proliferation and metastasis of CRC cells via Wnt/β-catenin signalling-mediated deubiquitination [127]. SMYD2 overexpression promoted cell proliferation but also increased the metastatic ability of CRC and may activate the Wnt/β-catenin pathway, as SMYD2 overexpression suppresses the expression of adenomatous polyposis coli 2 (APC2), an inhibitor of the Wnt/β-catenin pathway. The low APC2 expression in CRC cells was due to the modification of SMYD2-mediated DNA methylation [128]. TM4SF1 promoted epithelial–mesenchymal transition (EMT) and cancer stemness via the Wnt/β-catenin/SOX2 pathway in CRC, which revealed that TM4SF1 modulated SOX2 expression in a Wnt/β-catenin activation-dependent manner [129]. Mechanistically, WDR74 decreased the phosphorylation of β-catenin and induced the nuclear accumulation of β-catenin, activating the Wnt/β-catenin signalling pathway in CRC cells [130]. B7-H4 overexpression promoted the proliferation and invasion of CRC cells, which may be actioned by Wnt signalling. FOXS1 could also function as an oncogene, promoting CRC cell proliferation, migration, invasion, and metastasis through the Wnt/β-catenin signalling pathway [131]. Overexpressed FOXO4 inhibited the migration and metastasis of CRC cells by enhancing the APC2/β-catenin axis, suggesting that FOXO4 is a potential therapeutic target of CRC [132]. FUBP1 is an oncogene that initiates the development of CSCs by activating Wnt/β-catenin signalling via direct binding to the promoter of Dvl1, resulting in increased pluripotent transcription factors, including c-Myc, NANOG, and SOX2 [133].

Metabolic reprogramming is a hallmark of cancer cells, supporting their growth and metastasis [134,135]. The metastasis of cancer cells to distant organs requires the ability to adapt to metabolic conditions distinct from those at the primary tumour site. C14orf159, a d-amino acid metabolising enzyme, is located in the mitochondrial matrix to maintain the mitochondrial membrane potential and has the function of reducing the expression of genes involved in CRC metastasis. In CRC, C14orf159 is involved in Wnt6 expression and β-catenin activation via regulation of the mitochondrial membrane potential [136,137,138].

EMT is a critical process triggered during tumour metastasis, by which tumour cells convert from epithelial cells to mesenchymal cells, losing cell polarity and the ability for cell adhesion, allowing cancer cells to easily spread into the lymphatic system and blood circulation, invade surrounding tissues and migrate to distant organs, primarily including the liver and lung [139,140,141]. Elevated secretion levels of Wnt7b were significantly associated with lymphatic and remote metastasis and predicted a poor outcome in patients with CRC. Researchers found that either a partial knockdown of Wnt7b or a blockade of the Wnt/β-catenin signalling pathway reversed the EMT process and inhibited the migration of CRC cells [142]. Neuroligin 1 (NLGN1) promoted APC localisation to the cell membrane, and co-immunoprecipitates with some isoforms of this protein stimulated β-catenin translocation to the nucleus, upregulating mesenchymal markers and Wnt target genes and inducing an “EMT phenotype” in CRC cell lines [143]. Claudin-7 (Cldn7), a tight junction protein, was recently reported to function as a candidate tumour suppressor gene in CRC. Cldn7 deficiency conferred stemness properties in CRC through SOX9-mediated Wnt/β-catenin signalling to promote cellular EMT [144]. Chemotherapeutic resistance is the key characteristic of CSCs and contributes to postoperative recurrence and metastasis. The activation of LRP5 could affect the stem-like properties of CRC cells by activating the canonical Wnt/β-catenin pathway. Activation of LRP5 induced the drug resistance of CRC cells to platinum-based drugs [145].

Wnt/PCP signalling components are frequently dysregulated in solid tumours, and aberrant pathway activation contributes to tumour cell migratory properties [146]. Wnt protein is secreted on the surface of exosomes [147]. Wnt1 expression on CRC cell–derived exosomes was considerable and exosomal Wnt1 substantially enhanced the capacity for CRC cell proliferation and migration by activating the factors RHO, JNK, both of the downstream of Wnt/PCP signalling pathway [148]. Wnt2 involved in invasive activity of CRC cells through Wnt/PCP signalling pathway coupled to GSK3β and c-Jun/AP-1 signalling [149]. Meanwhile, overexpression of Wnt5b also induced the proliferation, migration, and invasion of CRC cells by activating the non-canonical Wnt/JNK signalling [150]. Wnt11 signalling diminished the E-cadherin-mediated cell–cell contact signalling resulting from stimulates proliferation and promotes morphological transformation in the intestinal epithelial cells, via enhancing the activation of protein kinase C and Ca^2+^/CaMKII [151]. FZD7 initiates both canonical and noncanonical Wnt signalling pathways to control cell motility, metastasis, and invasion. Significantly, Wnt11/FZD7 receptors binding activated Wnt/PCP signalling pathway to promote CRC cell proliferation, migration, and invasion activities through phosphorylation of JNK and c-Jun [152,153,154]. 

The downstream genes involved EMT express by activating Wnt/β-catenin pathway. The Wnt/PCP signalling pathway plays key roles of in guiding cell polarity and tumour cell invasiveness. 

## 7. The Wnt Signalling Pathway in CRC Prognosis

### 7.1. The Wnt Signalling Pathway in Chemotherapy Drug Resistance 

Chemotherapy is common for the treatment of CRC; among the most used chemotherapeutic drugs, approved as first- and second-line adjuvants in CRC, are 5-fluorouracil (5-FU), irinotecan (CPT-11), oxaliplatin capecitabine, and leucovorin [155]. For high-level microsatellite instability/deficient mismatch repair metastatic CRC, chemotherapy is also used in combination with immune checkpoint inhibitors, such as bevacizumab or cetuximab [156]. However, the treatment of patients with advanced grades of metastatic disease is unsatisfactory, mainly due to chemoresistance. Membrane transporters including pump P-glycoprotein (P-gp), the ATP-binding cassette (ABC) superfamily, and the multidrug resistance-associated protein (MRP) subfamily, induce the ability of tumour cells to efflux drugs, which could pump out chemotherapeutic agents, resulting in drug resistance [157]. CSC-mediated multidrug resistance, an altered TME, and the functions of some noncoding RNAs are also associated with chemoresistance in CRC [158]. Meanwhile, it has increasingly been known that the Wnt signalling pathway contributes to cancer drug resistance [159]. 

CRC stem cells (CRC-SCs) have some unique features, such as a high level of heterogeneity and plasticity, and also participate in drug resistance and cancer recurrence. During the treatment of patients with CRC, the genetic contents of SCs and niche of SCs aberrantly activate Wnt signalling pathways [160], leading to the accruing of CRC-SCs that are resistant to chemotherapy and radiotherapy. Side population cells (SP), as a group of CSCs, have obvious worldwide regard for their association with tumorigenesis and drug resistance, including CRC [161]. The protein tyrosine phosphatase receptor type C (PTPRC) encoded CD45, which correlates with the stemness of CRC cells, was increased in remnant tumour tissues after expression in both 5-FU-treated and irradiated CRC cells. CD45 binds only to degradable β-catenin and not to active β-catenin, reducing the tyrosine phosphorylation levels of degradable β-catenin by colocalisation with only total and degradable β-catenin at the cellular membrane, which collectively enhances stemness and the therapy-resistant phenotype [162]. Forkhead Box M1 (FOXM1) also participated in the malignant behaviours of CRC in which the Wnt/β-catenin signalling pathway is involved. FOXM1 bound to Dvl2 and enhanced the nuclear translocation of Dvl2 and the Dvl2-mediated transcriptional activity of Wnt/β-catenin, known to induce metastasis and drug resistance [163]. RAI2, an independent poor prognostic marker, inhibited Wnt signalling by interacting with or downregulating CtBP2, resulting in the repression of stem cell-like properties and the increased chemosensitivity of CRC cells to oxaliplatin and fluorouracil [164]. LGR5, a marker of CSCs, is a target gene of the Wnt pathway [130]. LGR5-positive CSCs are chemotherapy-resistant, making the cells enter a static state to escape drug treatment [165]. Activation of the LRP5 gene promotes CSC-like phenotypes, including tumorigenicity and platinum-based drug resistance in CRC cells, through activation of the canonical Wnt/β-catenin and IL-6/STAT3 signalling pathways [145]. The drug efflux effect of ABC transporters induces chemical resistance in numerous solid tumours [166]. Studies have shown that Wnt/β-catenin signalling is closely related to the ABC transporter of CSCs [167]; for example, the pivotal role of the c-Myc-ABCB5 axis in 5-FU resistance in CRC cell [168]. SP cells showed more resistance to 5-FU and irinotecan than non-SP cells, as well as a higher activation of the Wnt signalling pathway; inhibiting β-catenin significantly decreased SP [169]. The author found that silencing the SNTB1 expression involved in the Wnt/β-catenin pathway resulted in the downregulation of several β-catenin downstream proteins, such as c-Myc and cell cycle modulator CCND1, and, importantly, decreased the percentage and clonogenicity of SP cells [170]. 

The TME is a “complex network” of different cell types, various cytokine signalling molecules, and ECM components [171]. Wnt signalling pathway crosstalk between cancer cells and the surrounding supportive stromal cells in the TME promotes drug resistance [100]. CAFs are a key component of the TME and contribute to drug resistance, interacting with Wnt/β-catenin signalling [172,173,174]. Clear colocalisation between CAFs and tumour cells expressing nuclear β-catenin is observed in primary CRC samples. CAF-derived exosomes contribute to the secretion and carrying of Wnt ligands to activate Wnt/β-catenin signalling to regulate Wnt activity, resulting in the function of CSC characteristics being promoted [175]. Wnt16B and stable free radical polymerisation 2 (SFRP2) were produced by CAFs treated with chemotherapy, which synergistically activated the canonical Wnt pathway in cancer cells to reduce treatment sensitivity [176,177]. In hypoxia, hypoxia-inducible factor-1α/2α leads to the upregulation of the key Wnt coactivator BCL-9, and this crosstalk synergistically acts on the development of CRC resistance [178].

Metabolic reprogramming is one of the important hallmarks of drug resistance. Recent research found the effect of glucose metabolic reprogramming events of 5-FU resistance in CRC. The upregulation of HIF-1α, classical signalling the state of oxygen deficiency, promotes metabolic reprogramming that facilitates 5-FU resistance in CRC via reactive oxygen species (ROS) and the Wnt/β-catenin signalling pathway, instead of in the classical ways according to external oxygen concentrations [179]. Sec62 expression was upregulated by the m(6)A-mediated stabilisation of Sec62 mRNA, which restrains the sensitivity of CRC cells to chemotherapeutic drugs. Sec62 promoted the stemness of CRC cells through the activation of Wnt/β-catenin signalling. Mechanistically, Sec62 can bind to β-catenin and competitively disrupt the interaction between β-catenin and APC to inhibit the assembly of the β-catenin destruction complex, destroying the degradation of β-catenin [180]. Carnitine palmitoyltransferase 2 (CPT2) was decreased in CRC, and CPT2 downregulation could trigger stemness and oxaliplatin resistance in CRC via the activation of ROS/Wnt/β-catenin-induced glycolytic metabolism [181].

It has been increasingly recognised that long non-coding RNA (lncRNA, a non-coding regulatory RNA with greater than 200 nucleotides) regulates the Wnt/β-catenin pathway in multidrug-resistant CRC. Small nucleolar RNAs (snoRNAs) are a subset of nuclear non-coding (nc)RNAs with lengths of 60 to 300 nucleotides. The C/D box snoRNA SNORD1C was highly expressed in CRC to enhance cancer cell stemness and 5-FU resistance via the Wnt/β-catenin pathway, resulting in downregulation of c-Myc downstream, as well as other stem cell regulatory genes, such as CD44 and SOX2 [182].

### 7.2. The Wnt Signalling Pathway in Immunotherapy Drug Resistance

In recent times, immunotherapy has shown promising success in enhancing the treatment responses in some CRCs [183]. For example, nivolumab and pembmlizumab (Programmed cell death-1 blocking antibodies) have shown efficacy in the mismatch repair deficient high microsatellite instability (dMMR-MSI-H) subtype of metastatic CRC patients [183]. Immune checkpoint inhibitors such as nivolumab and pembrolizumab could reverse T cell dysfunction and apoptosis by inhibiting the effects of the interaction of programmed death-1 (PD-1) and its ligand PD-L1 to enhance T cell activation and cytotoxicity to tumour cells [184,185]. Moreover, chimeric antigen receptor-modified T (CAR-T) cells or cancer vaccines have also contributed to the development of immunotherapy for CRC [186]. However, only about 20% of patients respond to immune checkpoint blockade. Importantly, even if patients derived benefit from this treatment, the effect was not satisfactory due to developing resistance [187,188]. Preclinical and clinical models demonstrate that WNT/β-catenin activation leads to the suppression of CD8^+^ T-cell tumour infiltration and the evasion of immune elimination, which has been linked to decreased chemokine levels [189,190,191]. β-catenin could suppress the transcription of T-cell recruiting chemokine CCL4 from dendritic cells, which reduced the infiltration of CD8^+^ T cells within the TME in colon cancer [192]. Moreover, when WNT/β-catenin aberrantly activates, the recruitment of T cells to the TME is impaired, as a result of the failing recruitment of specific dendritic cells into the tumour leading to a lack of tumour-infiltrating T cells [193,194]. Mutations in components of the WNT/β-catenin signalling pathway induced downstream MYC expression and controlled PD-L1 and CD47 transcription, disfavouring the accumulation of tumour-associated T cells and macrophages [195]. Fermitin family member 3 (FERMT3) promoted the blockage of PD-L1 by FERMT3-mediated Wnt/β-catenin signalling to suppress CRC cell invasion, 5-FU resistance and NK cells-mediated tumour killing [196]. 

### 7.3. The Wnt Signalling Pathway as a Prognostic Indicator of CRC

The Wnt/β-catenin pathway plays an important role in advanced colorectal carcinoma [197]. Recently, a meta-analysis was performed to assess the prognostic value of β-catenin expression in CRC patients. High nuclear β-catenin expression was significantly associated with poorer disease-free survival (DFS), cancer-specific survival (CSS), and overall survival (OS) in patients with CRC whereas, low membranous β-catenin expression was associated with poor OS [198]. In univariate analysis, TCF4 expression turned out to be a negative prognostic factor being associated with shorter overall survival, whereas LEF1 expression as well as a LEF1/TCF4 ratio were positive prognostic factors and correlated with longer overall survival [199]. 

According to clinical, molecular, and pathologic features, mutations of F-box and WD repeat domain-containing 7 (FBXW7), a member of the Dickkopf (DKK) family, result in aberrant cytoplasmic and nuclear β-catenin accumulation, although at a lower rate than CTNNB1-mutated CRC [200]. S100A4, a Wnt/β-catenin target gene, is an established prognostic biomarker for CRC patient survival. The cross-regulation of S100A4 with Wnt pathway antagonist Dickkopf-1 (DKK1) resulted in S100A4 overexpression downregulating DKK1, while S100A4 knock-down increased DKK1. Significantly, S100A4 can be seen as the predominant factor in this feedback loop in Wnt signalling modulation. The combination of S100A4 and DKK1 improved the identification of CRC patients at high risk and improved the prognosis of overall and metastasis-free survival [201]. Overexpression of UBE2M in CRC specimens contributed to a decreased overall survival of patients and mediated 5-FU and oxaliplatin resistance in CRC cells via the Wnt/β-catenin signalling pathway [202]. Paired-like homeodomain transcription factor 2 (PITX2) could act as an effective biomarker in human cancer diagnosis and prognosis progression of CRC; PITX2 enhanced resistance to 5-FU in CRC, upregulating the Wnt/β-catenin axis [203].

It is known that the Wnt signalling pathway contributes to cancer drug resistance. Wnt signalling pathway could also interfere with immunotherapy. Moreover, prognostic factors associated with Wnt signalling are also noteworthy.

## 8. Challenges and Prospects 

A large number of aspects of the Wnt pathway have been studied in CRC, the roles of therapeutic targets for CRC have been previously established. However, these extensive studies suggest that the regulation of Wnt signalling is more complicated than previously hypothesized, and thus various information still needs to be further clarified. Some colorectal tumours exhibit a reverse correlation between Wnt signalling levels and patient survival, due to the variability in signalling levels and driver mutations across different tissues [204], as well as the complexity of components involved in this pathway, including β-catenin, Axin, APC, GSK3β, Dvl, CK I, and the TCF/LEF family, in addition Wnt signalling play important role in homeostasis, which cause obstacles in developing Wnt-suppressing therapy [205]. To date, there are no FDA-approved Wnt signalling inhibitors in clinical use.

Admittedly, with the development of targeted drugs and combination therapy strategies, many efforts have been made over the years to targeted intervention of Wnt signal transduction in CRC. In addition, several inhibitors have entered clinical studies [206]. For example, OMP-131R10 has been shown to inhibit the growth and metastasis of advanced CRC in phase I clinical trials [207].

CRC heterogeneity may render the difficulty to improve the rates of patients surviving. Thus, further insight into Wnt signalling pathway may enable deeper understanding of CRC to solve the clinical problems of tumour metastasis, recurrence and chemoresistance. Therefore, further studies may focus on the following: (i) Identifying the connection between Wnt signalling pathway and immunotherapy may be a promising therapeutic strategy for CRC. (ii) Researching the feasibility of certain multitarget therapeutic strategies concurrently may improve treatment. (iii) Understanding the crosstalk among oncogenes may cope with the complexity of CRC. (iv) Finding more information about components of Wnt signalling may provide opportunities to research the highly selective inhibitor for CRC to avoid unnecessary side effects.

## 9. Conclusions

The Wnt/β-catenin pathway affects almost all physiological and biochemical process. The Wnt1 and Wnt2 usually involved in Wnt/PCP signalling pathway, which plays key roles in guiding cell polarity and tumour cell invasiveness. Wnt5 and Wnt11 can have both oncogenic and tumour-suppressive functions. Their tumour-suppressing activities are mainly attributed to the activation of Wnt/Ca^2+^ signalling that may inhibit the Wnt/β-catenin signalling. Over the years, the molecular mechanisms and functional effects of the Wnt signalling pathway have been accumulatively studied at all stages of CRC (Figure 3). However, improving the cure rate and extended survival of CRC remains a challenge. Nevertheless, owing to the complexity of CRC, numerous details remain to be uncovered with regard to its connections to the Wnt signalling pathway, which would provide clearer guidance in individualised treatment and promote the development of specific inhibitors or drug combinations and prognosis markers to improve anticancer efficacy. It is hoped that immunotherapy can eradicate cancer. For CRC, the treatment response of patients remains unsatisfactory due to the differences between individuals.

## Figures and Tables

**Figure 1 cells-12-00447-f001:**
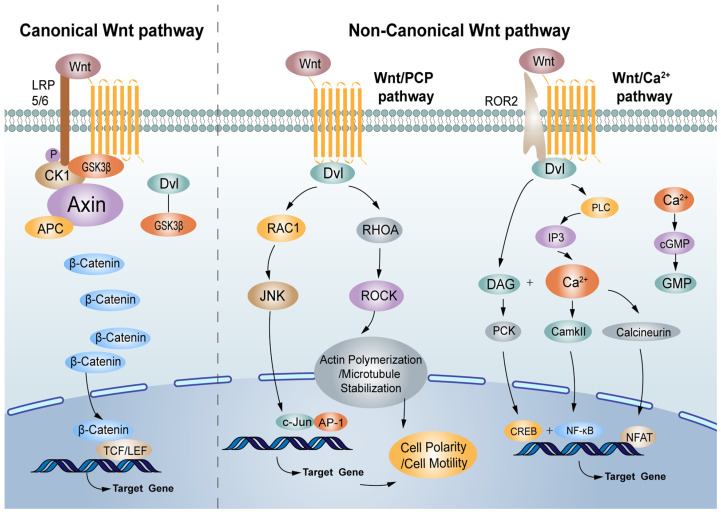
The canonical and non-canonical Wnt signalling pathway.

**Figure 2 cells-12-00447-f002:**
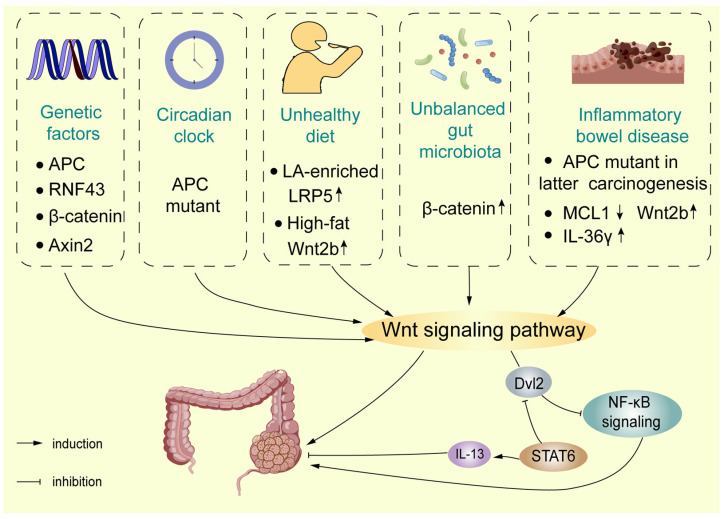
The Wnt signaling pathway in occurrence of colorectal cancer. Genetic factors, and environmental factors, such as unhealthy lifestyle, and imbalanced gut microbiota may induce colorectal cancer (CRC) by Wnt signaling pathway. Moreover, aberrant Wnt signaling pathway also play a crucial role colitis-associated carcinoma.

**Figure 3 cells-12-00447-f003:**
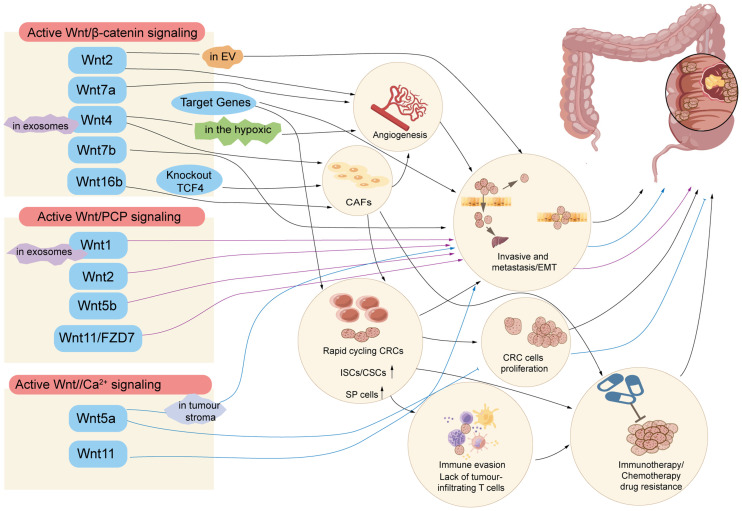
The Wnt signalling pathway in the biochemical process of colorectal cancer (CRC). The aberrant Wnt pathway drives the cancer stem cells, the tumour microenvironment, angiogenesis, invasion, metastasis, and drug resistance in CRC.

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
