# Peer review of "Advances of Wnt Signalling Pathway in Colorectal Cancer"

_cells, 2023, doi:10.3390/cells12030447_

Round 1

Reviewer 1 Report

The MS entitled “Advances of Wnt signalling pathway in colorectal cancer” by Zhu Y. and Li X. summarizes the most recent discoveries on Wnt signaling in the colorectal cancer context. The MS is well-written and contains a few relevant insights. However, some details need to be addressed prior to publication.

1.    In the sentence “The LRP5/6 receptors recruit the protein destruction complex and are phosphorylated by CK I and GSK3β, and Dishevelled (Dvl) is activated, which inhibits GSK3β[18],” it seems that Dishevelled works downstream to LRP5/6. However, it is common knowledge that Dishevelled is required for the phosphorylation of LRP5/6 PPPSPxS motifs. For this reason, Dishevelled is upstream to LRP5/6 phosphorylation.

2.    Please mention TCF/LEF as the transcription factors of the Wnt/β-catenin signaling. Specifically, TCF7, TCF7L1, TCF7L2 and LEF1, also known as TCF1, TCF3, TCF4 and LEF1 in the Wnt field. These transcription factors are often confused with other TCFs.

3.    “The Wnt signalling pathway is one of the most representative signalling pathways in CRC, playing an important role in the occurrence of CRC (Fig. 2).” How representative? Please add the expected % of CRC that has Wnt signaling mutations (e.g., TCGA data).

4.    What is the APC mutant GEMM? Is it a Apc Het model?

5.    There is not enough information regarding APC as a CRC model, how it has a high Wnt activity, and how frequent APC truncation is in human CRC.

6.    Section 4.2 is too brief.

7.    “Negative Wnt regulators Axin2 and ring finger protein 43 (RNF43) were strongly downregulated in IBD-CRCs, while polymeric immunoglobulin receptor (PIGR) and Oncostatin M Receptor (OSMR), which are involved in mucosal immunity, were dysregulated via epigenetic modifications.”
Although Axin2 and RNF43 are negative regulators of Wnt signaling, they are also known as Wnt signaling target genes. Does this data suggest that Wnt signaling is suppressed in IBD-CRCs?

8.    “In hypoxia, Wnt4-loaded exosomes secreted by the tumour cells via activated Wnt/β-catenin signalling promoted angiogenesis through the proliferation and migration of ECs [88,101].”
How can Wnt4-loaded exosomes activate the Wnt signaling pathway? Considering that Wnt4 is an extracellular ligand. Are the exosomes loaded with Wnt4 protein or Wnt4 mRNA? Please provide further insides regarding Wnt and exosomes in the MS.

9.    A small conclusion of each section would improve the understanding of the review.

10. The review lacks insights related to clinical trials and insights related to the future of the CRC/Wnt field.

11. Figure 2 and Figure 3 are not helpful for the understanding of the MS.

12. In the abstract, it is highlighted that the review will display the impact of the “three branches” of the Wnt signaling in the CRC context. However, throughout the MS, it is hard for a reader to understand whether the novel finding is related to the canonical or the non-canonical pathway. For example, the term PCP is only used during the introduction of the MS, but not during the actual review.

13. Overall, the review is shallow. If possible, a more in-depth explanation of the papers would be helpful. Also, some insights connect the sections. The non-canonical part of the MS should be improved since most of the papers reviewed are related to the canonical pathway.

Reviewer 2 Report

The review article by Zhu and Li focusses on WNT signaling in colon cancer. I have the following specific comments.

- The description of canonical and non-canonical WNT signaling is quite technical on the signaling level. It would be nice if the authors could add some insight, what the function difference between these WNT signaling pathways is and what is when activated and impacts epithelial- or tissue homeostasis.

- There appears to be some redundancy/duplication between part 3 (lines 56-58) and part 4 (lines 98-100) in the description of factors that are involved in canonical and non-canonical WNT signaling. I would suggest to adjust this.

- In the section on colitis associated cancer it might be worth elaborating more on the molecular difference of CAC and sporadic CRC. CAC typically has early p53 mutations but does not appear to require APC mutations to initiate malignant transformation.

- The two sections about ulcerative colitis and inflammatory bowel disease are a little bit imprecise in clinical wording since IBD includes UC and Crohn’s disease. It would be nice to have some information about the specifics for each of these two IBDs.

- The info about beta-catenin mechanisms line 283-286 is a bit hard to understand. Consider rephrasing.

- In the section on WNT and cancer stem cells, it may be interesting to include studies on WNT-reporter based tumor initiation that found conflicting results.

- Please check consistency in spelling of proteins throughout the manuscript. E.g. WNT4 vs. Wnt4 (lines 330-340) but also applies to other factors.

- In the paragraph on EMT, I would suggest to elaborate on the relationship of WNT and typical EMT inducing factors, such as ZEB1 and Snail.

- Lines 375 and 486 read identical. Should be addressed.

- In the paragraph on WNT as a prognostic marker, it may be of interest to refer to studies that investigated WNT target genes and also beta-Catenin as prognostic markers.

Round 2

Reviewer 1 Report

The MS has been considerably improved. The authors have answered almost all of my questions. The MS is almost suitable for publication; just some minor changes should be fixed:

1. Figure 1. The model of non-canonical Wnt signaling doesn't suggest a role in cell migration. Note that most non-canonical Wnt models show the importance of actin polymerization and microtubule polymerization as a downstream effect. This should be included in Figure 1.

2. Minor typos should be fixed, such as "paly".

3. I still don't see a strong rationale to include Figure 2 and Figure 3 in the review. Figure 2 is a text. Also, the design is not attractive and very confusing. One example is pro-tumorigenic bacteria vs Gut microbiota. The drawing doesn't summarize the notion of pro-tumorigenic bacteria and gut microbiota. The same is true for Gene Mutation and Familial inheritance. Overall, Figure 2 confuses the reader and doesn't help the understanding of the MS. The same is valid for Figure 3.

Figure 3 suggests that the non-canonical pathway plays a major role in gene regulation. This ignores the very important role of Wnt-PCP in the regulation of cell migration. This is also a problem in Figure 1 and should be fixed. Another issue with Figure 3 is that the arrows go in both directions, which is extremely confusing. It seems to suggest that, for example, angiogenesis induces Wnt signaling, and angiogenesis induces "colorectal cancer". However, it does not indicate any role of Wnt signaling in colorectal cancer. The same is true for all the other examples. Therefore, I believe Figure 3 confuses the reader and doesn't help the understanding of the MS. Both Figure 2 and Figure 3 should be either improved or removed.

Author Response

Dear editor,

We greatly appreciate your valuable suggestions for our review manuscript entitled “Advances of Wnt signalling pathway in colorectal cancer”. We have revised our manuscript in accordance with the reviewers’ comments. Amendments are listed point-by-point in separate pages.

With these changes, we hope the Editorial Board find our revision satisfactory. We look forward to receiving your decision soon.

Sincerely yours,

Xia Li,Ph.D

Marine college, Shandong University; School of Pharmaceutical Sciences, Shandong University

Tel: +86-631-5688613; Fax: +86-531-88382012

E-mail :[email protected]

Response to Reviewer 1 Comments 

Point 1: Figure 1. The model of non-canonical Wnt signaling doesn't suggest a role in cell migration. Note that most non-canonical Wnt models show the importance of actin polymerization and microtubule polymerization as a downstream effect. This should be included in Figure 1.

 Response 1: Thanks for your good suggestion. The related information has been added in Figure 1. (Page 2)

Point 2: Minor typos should be fixed, such as "paly".

Response 2: We have checked carefully and corrected them as shown in revised version.

Point 3: I still don't see a strong rationale to include Figure 2 and Figure 3 in the review. Figure 2 is a text. Also, the design is not attractive and very confusing. One example is pro-tumorigenic bacteria vs Gut microbiota. The drawing doesn't summarize the notion of pro-tumorigenic bacteria and gut microbiota. The same is true for Gene Mutation and Familial inheritance. Overall, Figure 2 confuses the reader and doesn't help the understanding of the MS. The same is valid for Figure 3.

Figure 3 suggests that the non-canonical pathway plays a major role in gene regulation. This ignores the very important role of Wnt-PCP in the regulation of cell migration. This is also a problem in Figure 1 and should be fixed. Another issue with Figure 3 is that the arrows go in both directions, which is extremely confusing. It seems to suggest that, for example, angiogenesis induces Wnt signaling, and angiogenesis induces "colorectal cancer". However, it does not indicate any role of Wnt signaling in colorectal cancer. The same is true for all the other examples. Therefore, I believe Figure 3 confuses the reader and doesn't help the understanding of the MS. Both Figure 2 and Figure 3 should be either improved or removed.

Response 3: We have improved the Figure 2 and Figure 3 as shown in revised manuscript.

As shown in Page 4

The drawing statement of Figure 2 has been revised in MS. ( Page 4)

The Wnt signaling pathway in occurrence of colorectal cancer. Genetic factors, and environmental factors, such as unhealthy lifestyle, and imbalanced gut microbiota may induce colorectal cancer (CRC) by Wnt signaling pathway. Moreover, aberrant Wnt signaling pathway also play a crucial role colitis-associated carcinoma.

As shown in Page 15
